# Pan-Genome-Based Analysis as a Framework for Demarcating Two Closely Related Methanotroph Genera *Methylocystis* and *Methylosinus*

**DOI:** 10.3390/microorganisms8050768

**Published:** 2020-05-20

**Authors:** Igor Y. Oshkin, Kirill K. Miroshnikov, Denis S. Grouzdev, Svetlana N. Dedysh

**Affiliations:** 1Winogradsky Institute of Microbiology, Research Center of Biotechnology of the Russian Academy of Sciences, Moscow 119071, Russia; infon18@gmail.com (K.K.M.); dedysh@mail.ru (S.N.D.); 2Institute of Bioengineering, Research Center of Biotechnology of the Russian Academy of Sciences, Moscow 119071, Russia; denisgrouzdev@gmail.com

**Keywords:** methanotrophic bacteria, *Methylocystis*, *Methylosinus*, *Methylocystis heyeri*, complete genome sequence, pan-genome analysis, core genes, methane monooxygenase, motility genes, nitrogenase, environmental adaptations

## Abstract

The *Methylocystis* and *Methylosinus* are two of the five genera that were included in the first taxonomic framework of methanotrophic bacteria created half a century ago. Members of both genera are widely distributed in various environments and play a key role in reducing methane fluxes from soils and wetlands. The original separation of these methanotrophs in two distinct genera was based mainly on their differences in cell morphology. Further comparative studies that explored various single-gene-based phylogenies suggested the monophyletic nature of each of these genera. Current availability of genome sequences from members of the *Methylocystis*/*Methylosinus* clade opens the possibility for in-depth comparison of the genomic potentials of these methanotrophs. Here, we report the finished genome sequence of *Methylocystis* heyeri H2^T^ and compare it to 23 currently available genomes of *Methylocystis* and *Methylosinus* species. The phylogenomic analysis confirmed that members of these genera form two separate clades. The *Methylocystis*/*Methylosinus* pan-genome core comprised 1173 genes, with the accessory genome containing 4941 and 11,192 genes in the shell and the cloud, respectively. Major differences between the genome-encoded environmental traits of these methanotrophs include a variety of enzymes for methane oxidation and dinitrogen fixation as well as genomic determinants for cell motility and photosynthesis.

## 1. Introduction

Aerobic methane-oxidizing bacteria (methanotrophs) are able to utilize methane (CH_4_) as a sole source of energy [1,2,3,4]. The first step of CH_4_ oxidation is catalyzed by methane monooxygenase (MMO) enzymes, which exist in membrane-bound or particulate (pMMO) and soluble (sMMO) forms [5,6]. Described aerobic MOB include species of the *Gamma*- and *Alphaproteobacteria* as well as the *Verrucomicrobia* [7]. The latter are restricted to acidic geothermal habitats and grow as autotrophs, using CO_2_ as the carbon source via the Calvin cycle [8,9].

The *Methylocystis* and *Methylosinus* are two of the five genera that were included in the first taxonomic framework of methanotrophic bacteria created half a century ago [10]. These methanotrophs belong to the family *Methylocystaceae* of the class *Alphaproteobacteria* [11]. The initial separation of these methanotrophs in two different genera was based mainly on differences in their cell morphology and the type of resting cells. Members of the genus *Methylosinus* possess pyriform or vibrioid cells, which reproduce by binary and budding division and are usually arranged in rosettes [12]. In budding division, the bud contains a heat and desiccation-resistant exospore, which germinates into a vegetative daughter cell; this daughter cell is highly motile. By contrast, cells of *Methylocystis* species are rod-like to reniform in shape, occur singly, are non-motile, reproduce by binary division, and may form desiccation-resistant ‘lipid’ cysts as resting cells [13]. To date, the genus *Methylocystis* includes 6 species with validly published names, i.e., *Mc. parvus* [13], *Mc. echinoides* [14], *Mc. rosea* [15], *Mc. heyeri* [16], *Mc. hirsuta* [17], and *Mc. bryophila* [18]. The taxonomically described diversity within the genus *Methylosinus* is limited and includes two species only: *Ms. sporium* and *Ms. trichosporium* [12]. In addition to these described members, the genera *Methylocystis* and *Methylosinus* are represented by a number of taxonomically uncharacterized strains, which were isolated from a wide variety of habitats including freshwater sediments, upland soils, termite gut, paddy soils and wetlands [19,20,21,22]. Although cosmopolitan distribution is characteristic for members of both genera, cultivation-independent studies suggest *Methylocystis* species as one of the most abundant and functionally active methanotrophs groups in various terrestrial environments [23,24,25,26,27,28,29].

Early attempts to resolve the phylogeny of the *Methylocystis*/*Methylosinus* group based on 16S rRNA gene sequences and a limited number of described strains failed to reveal any clear clustering pattern [1,11,30]. Further examination of a large collection (~80 strains) of these methanotrophs isolated from diverse environments demonstrated that molecular phylogenies constructed from nearly complete 16S rRNA gene sequences and partial sequences of genes encoding PmoA (a subunit of pMMO) identified monophyletic clusters that could be correlated to three morphologically recognizable groups defined by *Methylosinus trichosporium*, *Methylosinus sporium* and *Methylocystis* spp. [20].

Growing number of available genome sequences from *Methylocystis* and *Methylosinus* species gives an opportunity to gain a deeper insight into the genome-encoded features of these methanotrophs. With the single exception of *Mc. echinoides*, genome sequences are now available for all described *Methylocystis* and *Methylosinus* species as well as for some representatives of these genera with as-yet-undefined taxonomic status. Genome analyses performed for representatives of both genera demonstrated the variability in the number and types of encoded methane monooxygenases. Besides conventional pMMO1 and soluble sMMO, these methanotrophs possess divergent methane monooxygenases pMMO2 and pXMO. pMMO2 was first discovered in *Methylocystis* sp. SC2 [21] and is associated with high affinity methane oxidation [31]. pXMO was shown to be present in many gammaproteobacterial methanotrophs [32] and is assumed to play a role in methane oxidation under hypoxic conditions [33].

Recently, we reported a draft genome sequence of *Mc. heyeri* H2^T^ [34]. In this work, we undertook additional sequencing effort to assemble the complete genome of this peat-inhabiting methanotrophic bacterium. We used this newly assembled genome as well as all available genome sequences from other representatives of the *Methylosinus*/*Methylocystis* group in order to re-examine their phylogeny by means of comparative genome analysis and to reveal the genome-encoded traits, which differentiate these closely related methanotrophs.

## 2. Materials and Methods

### 2.1. Cultivation Procedure and DNA Extraction

The culture of *Mc. heyeri* H2^T^ (= DSM 16984^T^ = VKM B-2426^T^) was grown in a liquid mineral medium M2, containing (in grams per liter of distilled water) KNO_3_, 0.2; KH_2_PO_4_, 0.04; MgSO_4_ × 7H_2_O, 0.02; and CaCl_2_ × 2H_2_O, 0.004, with the addition of 0.1% (by volume) of a trace elements stock solution containing (in grams per liter) EDTA, 5; FeSO_4_ × 7H_2_O, 2; ZnSO_4_ × 7H_2_O, 0.1; MnCl_2_ × 4H_2_O, 0.03; CoCl_2_ × 6H_2_O, 0.2; CuCl_2_ × 5H_2_O, 0.1; NiCl_2_ × 6H_2_O, 0.02; and Na_2_MoO_4_, 0.03. A set of 500 mL bottles were filled to 30% capacity with this medium, sealed with rubber septa, and CH_4_ (30%, v/v) was added to the headspace using syringes equipped with disposable filters (0.22 µm). The cells were harvested after incubation at 24 °C on a rotary shaker at 100 rpm for 5 days. DNA extraction was done as described by Oshkin et al. [34].

### 2.2. Genome Sequencing and Assembly

The complete genome of *Mc. heyeri* H2^T^ was assembled by combining a draft genome sequence of this methanotroph obtained in our previous study using PacBio RS II system [34] with the sequencing reads retrieved in this work using Illumina HiSeq 2500 platform (Illumina Inc., San Diego, CA, USA). The library was constructed with the NEBNext DNA library prep reagent set, according to the Illumina protocol. Raw sequence reads were quality checked with FastQC version 11.7 (https://www.bioinformatics.babraham.ac.uk/projects/fastqc/), and low-quality reads were trimmed using Trimmomatic v. 0.36 [35], with the default settings for paired-end reads. Hybrid assembly and circularization of PacBio and Illumina reads were performed with Unicycler v. 0.4.8.0 [36]. The finished genome sequence of *Mc. heyeri* H2^T^ has been deposited in the DDBJ/NCBI/EMBL databases (accession no. CP046052.1).

### 2.3. Selection of Reference Genomes and Phylogenomic Analysis

The available genome sequences of *Methylocystis* and *Methylosinus* species were retrieved from the GenBank (https://www.ncbi.nlm.nih.gov/genbank/). Of these, the complete genome sequences were available for *Methylocystis* sp. SC2 [37], *Mc. bryophila* S285 [38], *Mc. rosea* GW6, *Mc. rosea* BRCS1, *Mc. parvus* BRCS2, *Ms. trichosporium* OB3b^T^ [39] and *Methylosinus* sp. C49. The quality of other genomes was assessed using CheckM v.1.0.13 [40] to exclude genomes with less than 90% completeness. The resulting set of methanotroph genomes selected for the analysis included 13 genomes of *Methylocystis* species and 10 genomes from *Methylosinus* species (Table 1).

A genome-based tree for members of the *Methylocystis*/*Methylosinus* group was reconstructed using the Genome Taxonomy Database and GTDB-toolkit (https://github.com/Ecogenomics/GtdbTk) [41], release 04-RS89. The maximum likelihood phylogenetic tree was constructed using Mega X software [42].

Average nucleotide identities across *Methylocystis* and *Methylosinus* genomes were estimated using the program «Anvi-compute-genome-similarity» of the Anvi’o platform [43]. The resulting distance matrices were further visualized as a heatmap.

### 2.4. Core- and Pan-Genome Analysis

GET_HOMOLOGUES [44] software was used for the construction of pan-genome. BLASTp [45] hits with a minimum of 75% alignment length coverage and an E-value ≤1e-5 were considered. Clustering was performed based on OrthoMCL [46] algorithms with inflation parameter of 1.5. The pan-genome was divided into core, soft core, shell, and cloud genes. Core genes are defined as those present in all considered genomes, soft core genes are found in 95% of genomes, shell genes are present in more than 2 genomes and less than 95% of genomes, while cloud genes are found in not more than two genomes [45,46,47].

Additionally, the Anvi’o pangenomic workflow was used as a second method to construct the pan-genome [43,47]. The genomes were organized based on the distribution of gene clusters using MCL algorithm (Distance: Euclidean; Linkage: Ward). This method was also applied to determine singletons and single-copy gene clusters (SCGs). Singletons are defined as genes present only in single genomes. SCGs are represented by only one copy in each genome. Genes representing core, shell and cloud genome were functionally annotated against COG database [48] using eggNOG-mapper v2 [49]. Heatmaps based on the annotated COG functions of the core and singleton gene clusters were then plotted in R [50].

### 2.5. Analysis of Functional Genes

Comparison of selected genes encoding several environmentally relevant traits was performed using the BLASTp [51]. Phylogenetic clustering of the nitrogenase-encoding genes was visualized using the Cytoscape program [52]. The location of target genes in the genomes was analyzed using the UGENE software package [53]. Comparison of gene clusters was performed using the BLASTp algorithm, and the results were visualized using the Easyfig program [54]. Intra-genomic rearrangements were studied using the progressive Mauve algorithm [55].

## 3. Results

### 3.1. Genome Characteristics of Mc. heyeri H2^T^

The genome assembly obtained for *Mc. heyeri* H2^T^ in our previous study using PacBio RS II system consisted of 12 genomic contigs with an N50 value of 3,287,266 bp [34]. The additional round of sequencing on Illumina HiSeq 2500 platform made in this work yielded a total of 4,726,348 paired-end reads, with a mean read length of 150 bp. The final hybrid genome assembly for *Mc. heyeri* H2^T^ consisted of one circular chromosome sequence of 4,551,947 bp (G+C content of 63.1%) and two plasmids with lengths of 145,219 and 30,622 bp and G+C contents of 61.1 and 60.2%, respectively. The chromosome contains three identical *rrn* operon copies (16S-23S-5S rRNA), 53 tRNA genes, 3976 predicted protein-coding sequences, at least one Clustered Regularly Interspaced Short Palindromic Repeats (CRISPR) locus, and a set of CRISPR-associated (*cas*) genes. Relatively low number of insertion sequence elements (48) suggests moderate genome plasticity.

### 3.2. Genome-Based Phylogeny

The genome-based phylogeny of representatives of the genera *Methylosinus* and *Methylocystis* was determined based on the comparative sequence analysis of 120 ubiquitous single-copy proteins (Figure 1). Members of these two genera formed two separate clades which, in their turn, formed a super-clade of *Methylocystaceae* methanotrophs. The latter was clearly divergent from the clade of *Beijerinckiaceae* methanotrophs represented by members of the genera *Methylocella*, *Methylocapsa,* and *Methyloferula*. The species within the *Methylosinus* clade were clustered in 5 groups: I) *Ms.* sp. R-45379, *Ms.* sp. LW3, *Ms.* sp. C49; II) *Ms.* sp. PW1, *Ms.* sp. LW4; III) *Ms. sporium* SM89A, *Ms. sporium* DSM 17706^T^; IV) *Ms.* sp. Ce-a6; V) *Ms. trichosporium* OB3b^T^, *Ms.* sp. 3S-1. The average nucleotide identity (ANI) values were estimated for each pair of the genomes (Appendix A). ANI values calculated for four groups of *Methylosinus* species represented by several strains (I-III and V) were as follows: I) 89-94%; II) 90%; III) 95%; V) 97%. This indicates that *Ms. sporium* SM89A and *Ms. sporium* DSM 17706^T^ (group III) as well as *Ms. trichosporium* OB3b^T^ and *Ms.* sp. 3S-1 (group V) belong to the same species as intra-species level is defined at ≥95% ANI [56,57]. Thus, estimated diversity within the clade of currently available *Methylosinus* strains with determined genome sequences corresponds to 8 species.

The clade of *Methylocystis*-affiliated strains includes 6 groups: I) *Mc. heyeri* H2^T^, *Mc. bryophila* S285; II) *Mc. parvus* OBBP^T^, *Mc. parvus* BRCS2, *Mc.* sp. ATCC 49242; III) *Mc.* sp. B8; IV) *Mc.* sp. SC2; V) *Mc. hirsuta* CSC1^T^; VI) *Mc.* sp. MitZ-2018, *Mc. rosea* GW6, *Mc. rosea* SV97^T^, *Mc.* sp. SB2, *Mc. rosea* BRCS1. ANI values calculated for the groups represented by several strains were as follows: I) 68-69%; II) 78-97%; VI) 92-95%. Only *Mc. parvus* OBBP^T^ and *Mc. parvus* BRCS2 from group II as well as two sub-groups of strains from group VI (Mc. sp. MitZ-2018, Mc. sp. SB2 and *Mc. rosea* SV97^T^, Mc. sp. SB2, *Mc. rosea* BRCS1) can be classified as belonging to the same species. The total estimated diversity within the clade of analyzed *Methylocystis* strains, therefore, corresponds to 9 species.

### 3.3. Pan-Genome Analysis

To study the *Methylocystis*/*Methylosinus* pan-genome, two different approaches were used (see Methods). The first, GET_HOMOLOGUES-based approach clustered protein-coding sequences into core, soft core, shell, and cloud genomes [44]. Of 17,724 gene clusters, the *Methylocystis*/*Methylosinus* pan-genome core comprised 1173 genes (on average 29.3% of each genome), with the accessory genome containing 4941 gene clusters in the shell (27.9% of total gene clusters) and 11,192 in the cloud (63.1% of total gene clusters) (Figure 2).

The Anvi’o pangenomics workflow was used as a double-check procedure in order to verify the size of the core- and pan-genome. According to this approach, the *Methylocystis*/*Methylosinus* pan-genome comprised 18,322 gene clusters with 1341 genes in the core and 8065 gene clusters represented as singletons, which are defined as genes present in only one genome (Figure 3). Singletons constitute the cloud genome along with gene clusters containing genes from not more than two genomes. Although both approaches use MCL algorithm to identify clusters, Anvi’o pangenomics workflow detected more gene clusters. This possibly resulted in higher number of core genes. The pan-genome of *Methylocystis*/*Methylosinus* group is large and open. Appendix A displays the number of core genes (A) and the pan-genome size (B) as a function of the number of included genomes. Fitting the curve in Appendix A to a power law (4366.6n^0.2969^) allows extrapolating calculations to n genomes. Even if 100 genome sequences would have been determined, it is expected that sequencing another one would further add 50 genes to the pan-genome. Such calculations made separately for *Methylocystis* and *Methylosinus* pan-genomes resulted in 31 and 55 genes, reflecting less open pan-genome for the genus *Methylocystis* (Appendix A).

The core, shell, and cloud gene clusters were further annotated into COG classes (Figure 4). The core genome was mostly conserved in the following: translation and ribosomal biogenesis (10.5% of total core gene clusters), amino acid transport and metabolism (9.3%), energy production and conversion (8.3%), coenzyme transport and metabolism (6.9%), lipid transport and metabolism (6.9%), cell wall/membrane/envelope biogenesis (6.3%), and post-translational modifications (5.6%) (Figure 4). The classes for inorganic ion transport and metabolism (4.9%), replication, transcription, carbohydrate metabolism (4.4%), nucleotide transport and metabolism (3.3%), signal transduction (2.8%) and secondary metabolites biosynthesis, transport and catabolism (2.3%) were moderately abundant while intracellular trafficking and cell cycle control were limited in the core (1.5%).

The same functional categories, with addition of defense and cell motility functions, constituted the shell genome. High variability was observed for the transcription, replication and repair, cell wall/membrane/envelop biogenesis, inorganic ion transport and metabolism, and signal transduction. The latter category was especially well represented in the shell genome of *Methylosinus* strains (Figure 4). Indeed, the analysis of most abundant classes of signal transduction genes revealed that *Methylosinus* and *Methylocystis* genomes encode, respectively, 348 and 302 histidine kinases (HisKA, PF00512), 118 and 49 diguanylate cyclases (GGDEF domain, PF00990), 53 and 30 c-di-GMP-specific phosphodiesterases (EAL domain, PF00563), 89 and 7 methyl-accepting chemotaxis proteins (MCPsignal domain, PF00015). Overall, the total number of signaling proteins identified in shell genome constituted 658 for *Methylosinus* and 472 for *Methylocystis* strains. The most abundant functional categories in the cloud genome were represented by replication, transcription, cell wall/membrane/envelope biogenesis, and signal transduction mechanisms.

### 3.4. MMO-Encoding Genes

All methanotrophs of the *Methylocystis*/*Methylosinus* groups are characterized by the presence of conventional pMMO (or pMMO1) encoded by the *pmoCAB1* operon (Table 1). These genes are part of the core genome as they were found in all studied members of the *Methylocystis*/*Methylosinus* group. The *pmoCAB2* operon, which encodes high-affinity pMMO2, occurs in the genomes of most representatives of the genus *Methylosinus* except for *Ms. trichosporium* OB3b^T^ and *Methylosinus* sp. 3S-1. By contrast, *pmoCAB2* was identified in less than half of the analyzed *Methylocystis* genomes, including *Mc. heyeri* H2^T^, *Mc. parvus* OBBP^T^, *Mc. parvus* BRCS2, *Mc. bryophila* S285, *Methylocystis* sp. SC2 and *Methylocystis* sp. B8. An opposite distribution pattern was observed for the *pxmABC* operon, which was identified in genomes of seven *Methylocystis* strains and only one genome of *Methylosinus* species, i.e., *Methylosinus* sp. R-45379. Genes encoding sMMO were present in the genomes of nearly all representatives of the genus *Methylosinus* with the only exception of *Methylosinus* sp. Ce-a6. The occurrence of sMMO in members of the genus *Methylocystis* was rarer, as the corresponding operons were found only in the genomes of *Mc. hirsuta* CSC1^T^, *Mc. heyeri* H2^T^ and *Mc. bryophila* S285. The entire set of genes for the alternative methane monooxygenases was clustered in the shell pan-genome. Additionally, singleton *pmoC* copies were distributed across core, shell, and cloud pan-genome.

### 3.5. Nitrogenase-Encoding Genes

Genes coding for the classical molybdenum–iron (Mo-Fe) nitrogenase were identified in all studied genomes of *Methylocystis* and *Methylosinus* species (Figure 5) and formed a separate cluster in the core genome. Interestingly, an additional gene cluster of rare vanadium–iron (V-Fe) nitrogenases was identified among the shell genes. V-Fe nitrogenases are present only in some *Methylocystis* species, such as *Mc. heyeri* H2^T^, *Mc. bryophila* S285 and both strains of *Mc. parvus*, but are absent in all representatives of the genus *Methylosinus* (Figure 5).

Comparative sequence analysis of the nitrogenase beta subunit proteins from proteobacterial methanotrophs revealed four phylogenetically distinct groups, three of which correspond to Mo-Fe nitrogenases and one is represented by V-Fe nitrogenases (Figure 6). Notably, proteins from *Methylocystis* and *Methylosinus* species are present in all these four groups. This suggests that representatives of the *Methylocystis/Methylosinus* groups received nitrogenase-encoding genes from different sources.

### 3.6. Genomic Determinants of Flagella-Based Motility

Genes encoding flagella biosynthesis were found in the genomes of all *Methylosinus* species (Figure 5). The occurrence of these genes in the genomes of *Methylocystis* species is rare. The complete set of flagella-encoding genes was identified only in the genomes of *Methylocystis* sp. ATCC 49242 and two strains of *Mc. parvus*.

### 3.7. Phototrophy-Related Genes

Genes necessary for the synthesis of bacteriochlorophyll a and b (*bchBCDEGHIJNLMNOPXYZ* and *acsF*), reaction center of the photosynthetic complex (*pufABCML* and *pucC*) and a number of carotenoids (*crtCDF*) are present in the genomes of all *Methylocystis rosea* strains as well as *Methylocystis* sp. MitZ-2018 and SB2 (Appendix A). Most of these genes are arranged in a single gene cluster (Figure 7), which is a part of the long conservative region (~195 kbp; locus 1487014-1681855 in the genome of *Methylocystis rosea* BRCS1) present in the genomes of all *Mc. rosea* representatives. This region also includes a number of genes associated with the nitrogenase complex, regulatory and transport genes.

## 4. Discussion

In this work, methanotrophs of the *Methylocystis*/*Methylosinus* group were studied using pan-genomic approach. We analyzed all publicly available, good-quality genome sequences of the taxonomically described *Methylocystis* and *Methylosinus* species and strains with as-yet-undefined taxonomic status. In addition, the complete genome sequence of *Mc. heyeri* H2^T^ was obtained and included in the pan-genomic workflow. Genome-based phylogenetic reconstructions and pair-wise ANI calculations performed in our study confirm the clear demarcation of these methanotroph genera, which was suggested earlier based on the comparative analysis of 16S rRNA and *pmoA* gene sequences [20].

The key features common to *Methylocystis* and *Methylosinus* species are the possession of an extended set of MMO-encoding genes and the ability to fix dinitrogen. All members of these genera possess conventional pMMO, pMMO1. One or two copies of the corresponding gene clusters (Table 1) are part of the core genome of these methanotrophs. The genes encoding all other currently known MMO types are clustered separately in the shell genome. With the only exception of Mc. sp. ATCC 49242, all members of the *Methylocystis*/*Methylosinus* group possess several MMO types. In addition to pMMO1, many *Methylocystis* and *Methylosinus* strains possess pMMO2, which is associated with high-affinity methane oxidation [31]. Although *pmoA*2 gene was first identified in *Methylocystis* sp. SC2 [21], the corresponding enzyme is encoded only in half of the examined *Methylocystis* genomes. In contrast, *pmoCAB2* operon was detected in the genomes of most *Methylosinus* strains. Overall, this indicates that significant part of organisms from *Methylosinus*/*Methylocystis* group has metabolic potential to thrive both in high-methane and low-methane environments. The *pxmABC* operon encoding pXMO was encoded in eight of 23 examined methanotroph genomes, including seven *Methylocystis* genomes and only one *Methylosinus* genome (*Methylosinus* sp. R-45379). Recent evidence for *pxmABC* expression in response to hypoxia suggests that pXMO is important for the survival of methanotrophs under limited O_2_ concentrations [33]. However, two *Methylosinus* strains included in this study, strains LW3 and LW4, were isolated from the Lake Washington sediment where concentration of oxygen is low and drops significantly within top 0.8 cm of the sediment [19]. No pXMO is encoded in their genomes although, apparently, they should be able to survive at low oxygen concentration. Finally, the presence of sMMO seems to be a characteristic feature of *Methylosinus* species since it was encoded in almost all of the examined *Methylosinus* genomes except of strain Ce-a6. By contrast, only three of the examined *Methylocystis* genomes encoded sMMO.

*Methylocystis* and *Methylosinus* species have long been known for their ability to fix atmospheric nitrogen [14], which is one of the most important environmental traits of methanotrophic bacteria. Indeed, genes encoding classical Mo-Fe nitrogenases were identified in all studied genomes. Interestingly, a separate cluster in the pan-genome was formed by the genes encoding alternative nitrogenase, which contains vanadium and iron as active site cofactors (V-Fe-nitrogenase). This type of nitrogenase was present in several strains of *Methylocystis* species only (Figure 5). The Mo- and V-containing nitrogenases are not equally distributed in nature [59]. While Mo-Fe-nitrogenases are found in all known diazotrophs, V-Fe-nitrogenases have been identified in the genomes of a limited number of prokaryotes [60]. These include some diazotrophs within the *Gamma*- and *Deltaproteobacteria*, *Cyanobacteria*, *Firmicutes*, and *Euryarchaeota* from a wide variety of habitats such as freshwater, lichens, termite gut, freshwater, and marine sediments, wheat roots and soils [61]. The search among sequenced genomes suggested that organisms with alternative nitrogenases are common in sediments. Recently, the presence of V-Fe-nitrogenase was reported for *Methylocystis bryophila* S285 [38] and *Methylospira mobilis* Shm1 [62]. We also searched for V-Fe-nitrogenase in other available methanotroph genomes. In addition to *Ms. mobilis* Shm1, *Mc. bryophila* S285, *Mc. parvus* BRCS2 and *Mc. parvus* OBBP^T^, *vnf* genes were present only in ‘*Ca*. Methylobacter oryzae’ [63]. Notably, all *Methylocystis* strains with V-Fe-nitrogenases were isolated from wetland ecosystems. The conventional Mo-Fe-nitrogenase was shown to be synthesized in the absence of a fixed nitrogen source when Mo is available, whereas the V-Fe-nitrogenase is expressed under Mo-deficient conditions in the presence of V [64]. We suggest that the presence of several forms of nitrogenases with different metals in the active center can be considered as an adaptation to the low availability of inorganic nutrients in wetlands. At the standard temperature for both in vivo and in vitro assays (i.e., 30 °C), the specific activity of V-Fe-nitrogenase is consistently lower than that of its Mo-Fe-counterpart. When the temperature is lowered (e.g., to 5 °C), V-Fe-nitrogenase becomes more active than Mo-Fe-nitrogenase [65]. Given that more than half of total wetland area is located between 50 and 70°N, the possession of V-Fe-nitrogenases might represent an adaptation strategy to low temperatures in these environments. Finally, the ability of V-Fe-nitrogenase to catalyze both CO and N_2_ reductions suggests a potential link between the evolution of the carbon and nitrogen cycles in wetlands [66].

One of the major features used to differentiate *Methylocystis* and *Methylosinus* species is the absence/presence of motility. Indeed, all currently characterized species of the genus *Methylocystis* were described as being represented by non-motile cells, while all *Methylosinus* species are characterized by flagella-based motility. This phenotypic differentiation was supported by the results of our analysis, which revealed the presence of flagella-encoding genes in all examined genomes of *Methylosinus* species. The complete sets of these genes were also revealed in the genomes of three *Methylocystis* strains (Figure 5). In addition, these strains also possessed histidine kinase CheA that mediates signal transduction in bacterial chemotaxis. However, we did not find any reports for their capacity to move. It may well be that expression of motility genes in these methanotrophs occurs in the environment but not under laboratory conditions. Experimental evidence for the presence of motility in *Methylocystis* species was obtained in the field study of Putkinen et al. [67] who demonstrated rapid, water-mediated colonization of *Sphagnum* mosses by peat-inhabiting members of the genus *Methylocystis*. The phenotypic difference between *Methylocystis* and *Methylosinus* species may be due to the different specialization of representatives of these genera. Indeed, the metabolic flexibility of *Methylocystis* species appears to be wider than that of *Methylosinus* species. Thus, many members of the genus *Methylocystis* are facultative methanotrophs capable of slow growth on acetate and ethanol in the absence of methane [68]. As shown in our study, some of these bacteria possess genome-encoded potential for phototrophy. Finally, possession of two different forms of nitrogenase also extends the range of environmental conditions suitable for growth of *Methylocystis* species. Thus, methanotrophs of the genus *Methylocystis* are potentially adapted to a wider range of environmental conditions than representatives of the genus *Methylosinus.* The latter, therefore, move to more favorable environment through flagella-associated motility. The composition of signal transduction proteins directly associated with chemotaxis also supports this idea. Those are mainly represented by MCPs, the predominant chemoreceptors in bacteria which are more abundant in genomes of motile microbes [69]. *Methylocystis* genomes encode very few MCPs, while the number of those present in *Methylosinus* strains is one order of magnitude higher.

Representatives of one particular species of the genus *Methylocystis*, *Mc. rosea*, are characterized by the presence of a gene cluster associated with phototrophy; these genes were not detected in the genomes of *Methylosinus* species. Similar genes have been previously found in some other alphaproteobacterial methanotrophs of the family *Beijerinckiaceae*, namely the species *Methylocapsa palsarum* and *Methylocella silvestris* [70]. In contrast to methanotrophic representatives of the *Beijerinckiaceae*, the organization of phototrophy-related gene clusters in *Mc. rosea* is extremely conservative (Figure 7). This may be due to the peculiarities of regulation of this gene cluster. The high similarity in the organization of genetic clusters may confirm the recently proposed hypothesis about the origin of phototrophy-related genes from a common ancestor [71]. According to this theory, the majority of the *Methylocystis* genus representatives have lost the gene cluster associated with phototrophy due to the specialization in methanotrophy.

In summary, the pan-genome analysis of 23 genome sequences from seven taxonomically described and 16 as-yet-uncharacterized *Methylocystis* and *Methylosinus* strains proved to be an effective tool for the demarcation of these genera and identification of diverse genome-encoded traits that determine adaptations of these methanotrophs to a wide variety of environments. Open pan-genome of the *Methylocystis*/*Methylosinus* group highlights the need for further efforts in extending the number of characterized representatives of these bacteria with determined genome sequences.

## Figures and Tables

**Figure 1 microorganisms-08-00768-f001:**
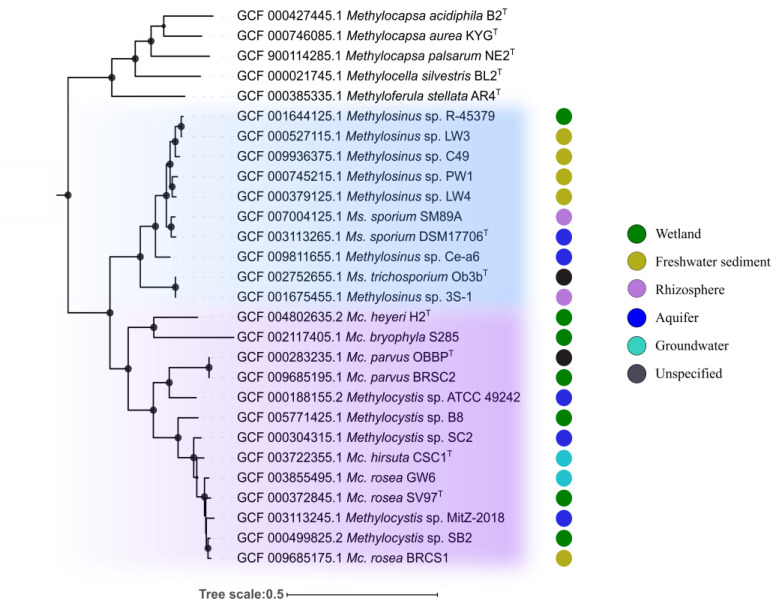
Phylogenomic tree showing the positions of representatives of the genera *Methylocystis* and *Methylosinus* in relation to other alphaproteobacterial methanotrophs based on the comparative sequence analysis of 120 ubiquitous single-copy proteins. The clades composed of *Methylosinus* and *Methylocystis* species are highlighted by blue and purple, respectively. The colored circles indicate isolation sources. The tree was constructed using the Genome Taxonomy Database toolkit [41], release 04-RS89. The significance levels of interior branch points obtained in maximum-likelihood analysis were determined by bootstrap analysis (100 data re-samplings). Bootstrap values of > 70% are shown by black circles. The root (not shown) is composed of 42 genomes of gammaproteobacterial methanotrophs. Scale bar indicates the number of substitutions per amino acid position.

**Figure 2 microorganisms-08-00768-f002:**
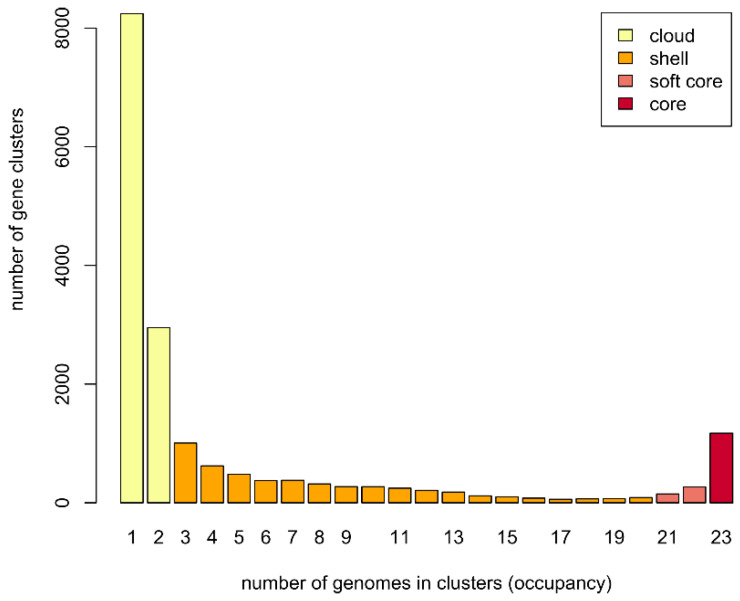
Barplot of the pan-genome matrix.

**Figure 3 microorganisms-08-00768-f003:**
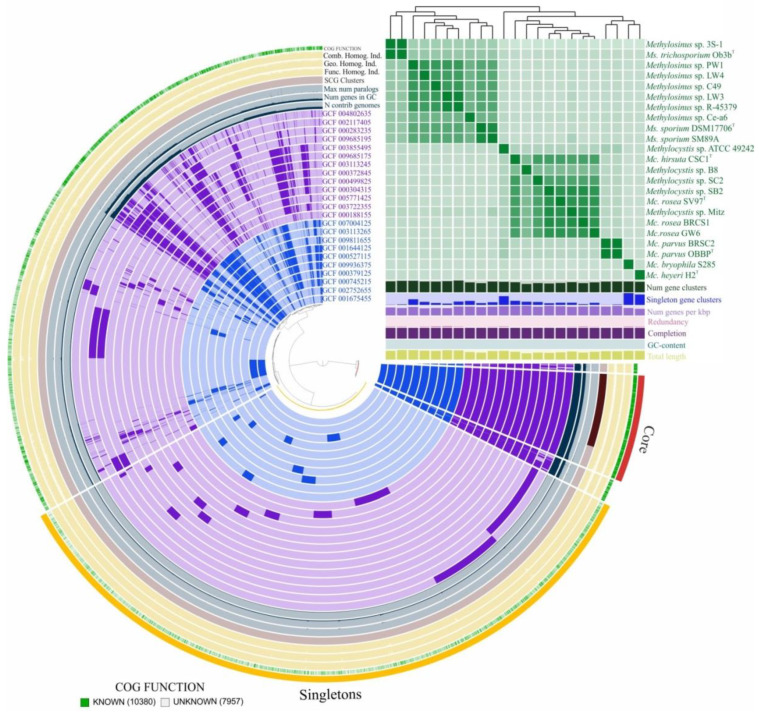
Pan-genome analysis. Clustering of genomes based on the presence/absence patterns of 18,322 pan-genomic clusters. The genomes are organized in radial layers as core, singleton (cloud) and accessory (shell and cloud) gene clusters [Euclidean distance; Ward linkage] which are defined by the gene tree in the center. Genomes of *Methylocystis* and *Methylosinus* species are colored by purple and blue, respectively. Line segments indicate single gene clusters. The outermost circles carry the information regarding the number of genomes containing specific gene cluster, number of genes in gene clusters, maximum number of paralogs (grey circles), distribution of single-copy gene clusters (pale pink circle), functional, geometric and combined homogeneity indexes (yellow). The layers below the heatmap display from the bottom to top: total length of the genomes, GC content, genomes completion, redundancy of each genome, number of genes per kbp, number of singleton gene clusters, total number of gene clusters. Heatmap denotes correlation between the genomes based on average nucleotide identity values calculated using pyANI [58].

**Figure 4 microorganisms-08-00768-f004:**
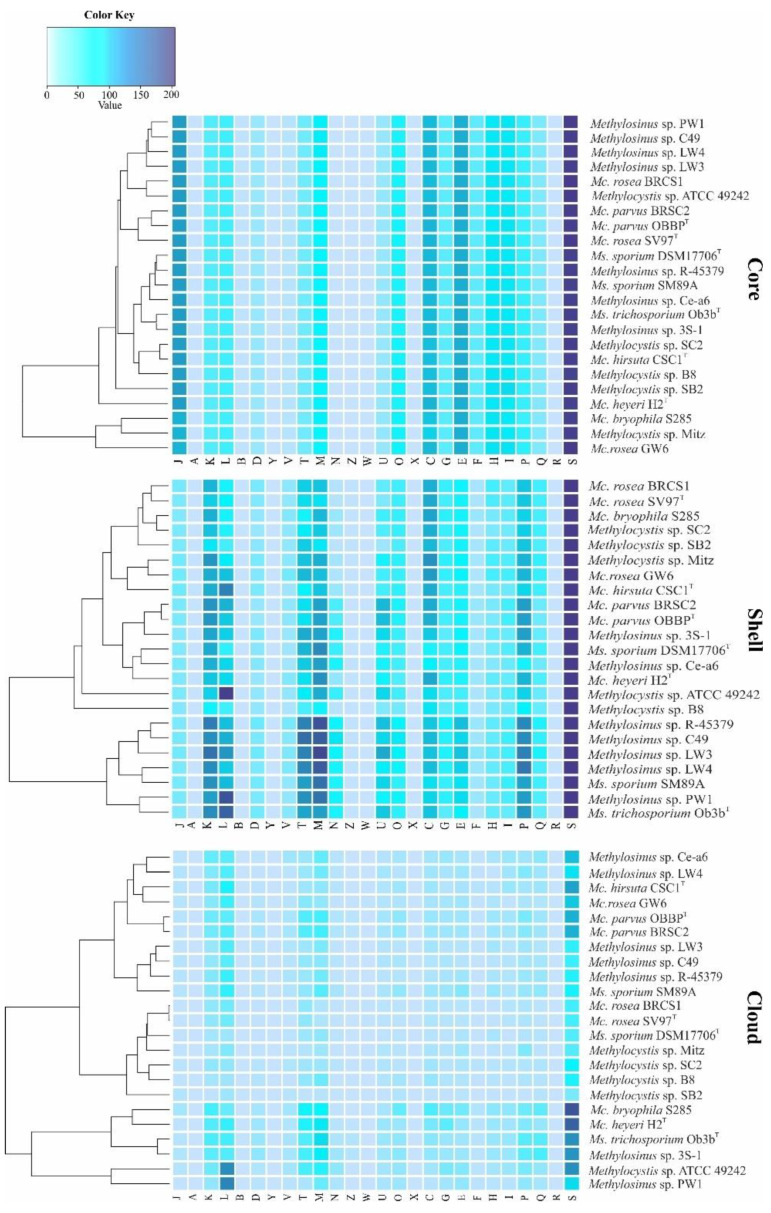
Functional annotation of core, shell, and cloud genes against COG database. Genes were automatically assigned to the following COG categories: A, RNA processing and modification; B, chromatin structure and dynamics; C, energy production and conversion; D, cell cycle control and mitosis; E, amino acid metabolism and transport; F, nucleotide metabolism and transport; G, carbohydrate metabolism and transport; H, coenzyme metabolism, I, lipid metabolism; J, translation; K, transcription; L, replication and repair; M, cell wall/membrane/envelop biogenesis; N, cell motility; O, post-translational modification, protein turnover, chaperone functions; P, inorganic ion transport and metabolism; Q, secondary structure; T, signal transduction; U, intracellular trafficking and secretion; Y, nuclear structure; V, defense mechanisms; Z, cytoskeleton; W, extracellular structures; X, mobilome: prophages, transposons; R, general functional prediction only; S, function unknown. The color key reflects the total number of genes in each COG category.

**Figure 5 microorganisms-08-00768-f005:**
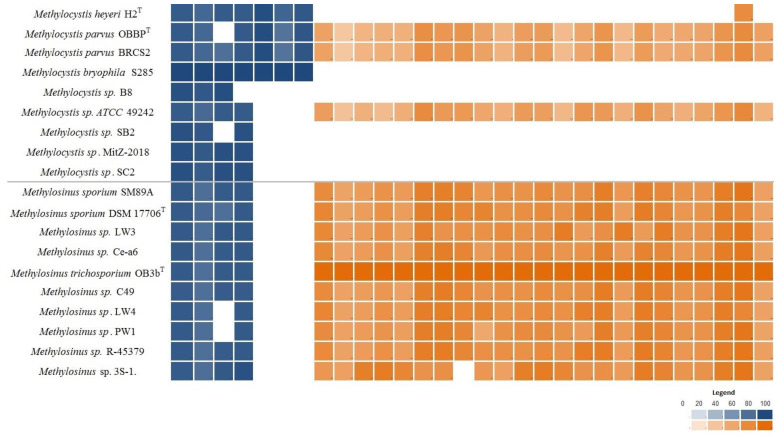
Heatmap displaying distribution of nitrogenase-encoding genes (shown in blue) and genes encoding flagella biosynthesis (shown in orange) in the genomes of *Methylocystis* and *Methylosinus* species. The color intensity reflects the percent identity shared with the reference proteins: nitrogenase complex proteins from *Mc. bryophila* S285 and motility-related genes from *Ms. trichosporium* OB3b^T^.

**Figure 6 microorganisms-08-00768-f006:**
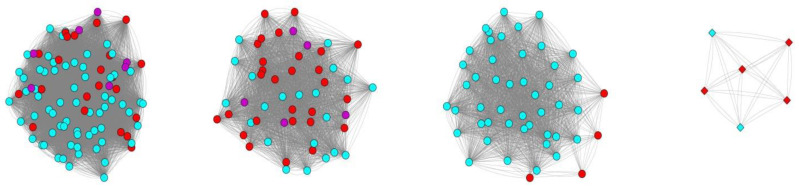
Comparison of proteins of nitrogenase beta subunit in methanotrophic bacteria. The nitrogenase complexes of methanotrophic bacteria clustered in 4 independent groups according to the level of amino acid identity. Edges denote amino acid identity of two proteins greater than 50%. Mo-Fe nitrogenases are indicated by circles, V-Fe nitrogenases—by diamonds. Nitrogenases from members of the family *Methylocystaceae* are colored in red; *Methylococcaceae*—in cyan; *Beijerinkiaceae*—in purple.

**Figure 7 microorganisms-08-00768-f007:**
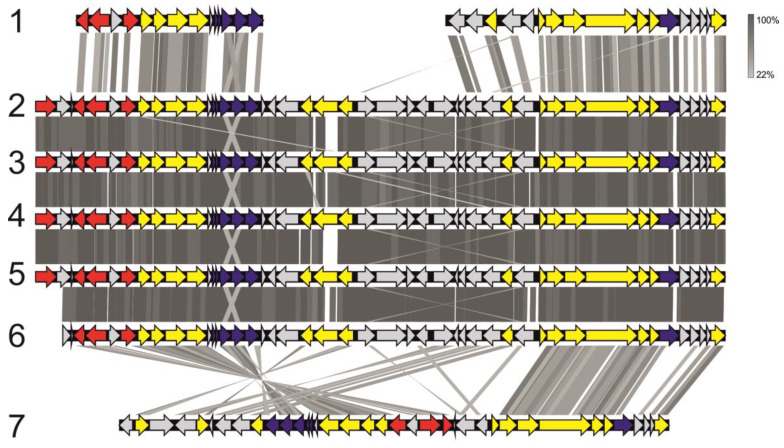
Organization of gene clusters associated with phototrophy in the genomes of *Methylocystis* species in comparison to that in the genomes of the phylogenetically-related methylo- and methanotrophs. The picture shows: *Methylobacterium phyllosphaerae* CBMB27 (1), *Mc. rosea* BRCS1 (2), *Mc. rosea* GW6 (3), *Mc. rosea* SV97^T^ (4) *Methylocystis* sp. Mitz-2018 (5), *Methylocystis* sp. SB2 (6) and *Methylocapsa palsarum* NE2^T^ (7). Grey strips reflect the identity level of homologous proteins (see color code). Genes encoding photosystem II, carotenoid biosynthesis, and bacteriochlorophyll synthesis are shown in dark purple, red, and yellow, respectively. Other genes are shown in light grey.

**Table 1 microorganisms-08-00768-t001:** General features of the genomes of *Methylocystis* and *Methylosinus* species selected for the analysis. *Methylocystis* and *Methylosinus* genomes are highlighted by purple and blue, respectively.

Group	Organism	Genome Assembly	Size (MB)	Contigs	G+C Content (mol %)	Proteins	rRNAs (5S, 16S, 23S)	tRNAs	pMMO1	pMMO2	sMMO	pXMO
*Methylocystis*	*Mc. heyeri* H2^T^	GCA_004802635.2	4.55	1	63.1	3976	3, 3, 3	53	2	1	1	none
	*Mc. bryophila* S285	GCA_002117405.1	4.53	1	63.2	4148	2, 2, 2	47	2	1	1	1
	*Mc.* sp. SC2	GCA_000304315.1	3.77	1	63.4	3545	1,1,1	46	2	1	none	none
	*Mc.* sp. ATCC 49242	GCA_000188155.3	4.73	7	62.8	4285	2,2,2	53	2	none	none	none
	*Mc.* sp. B8	GCA_005771425.1	3.41	28	61.2	3153	1, 1, 1	47	1	1	none	none
	*Mc.* sp. SB2	GCA_000499825.2	3.64	158	62.7	3392	1, 1, 1	45	2	none	none	1
	*Mc. rosea* SV97^T^	GCA_000372845.1	3.91	4	62.5	3639	1,1,1	48	2	none	none	1
	*Mc. rosea* GW6	GCA_003855495.1	3.64	1	62.8	3441	1,1,1	48	2	none	none	1
	*Mc. rosea* BRCS1	GCA_009685175.1	3.80	3	62.7	3499	1, 1, 1	49	2	none	none	1
	*Mc. parvus* BRCS2	GCA_009685195.1	4.53	1	63.4	4185	2,2,2	48	2	1	none	none
	*Mc. parvus* OBBP^T^	GCA_000283235.1	4.48	108	63.4	4128	1, 1, 1	46	1	1	none	none
	*Mc. hirsuta* CSC1^T^	GCA_003722355.1	4.21	4	62.4	4036	1, 1, 1	49	2	none	1	1
	*Mc.* sp. MitZ-2018	GCA_003113245.1	4.36	55	62.5	3962	1, 1, 1	50	2	none	none	1
*Methylosinus*	*Ms. sporium* SM89A	GCA_007004125.1	4.59	161	64.5	4155	1, 1, 1	46	1	1	2	none
	*Ms. sporium* DSM 17706^T^	GCA_003113265.1	3.8	55	65.2	3506	1, 1, 1	41	1	1	2	none
	*Ms trichosporium* OB3b^T^	GCA_002752655.1	4.5	1	65.8	4546	3, 3, 3	53	2	none	1	none
	*Ms.* sp. 3S-1	GCA_001675455.1	4.76	159	66.0	4209	1, 1, 1	48	1	none	1	none
	*Ms.* sp. PW1	GCA_000745215.1	5.13	12	64.7	4657	4, 4, 5	58	2	1	1	none
	*Ms.* sp. LW3	GCA_000527115.1	5.09	6	64.7	4502	3,3,3	52	2	1	1	none
	*Ms.* sp. LW4	GCA_000379125.1	4.82	16	64.9	4452	3,3,3	53	2	1	1	none
	*Ms.* sp. R-45379	GCA_001644125.1	4.97	319	64.4	4446	1, 1, 1	46	1	1	1	1
	*Ms.* sp. Ce-a6	GCA_009811655.1	4.09	119	65.4	3724	1, 1, 1	51	1	1	none	none
	*Ms.* sp. C49	GCA_009936375.1	4.71	1	64.9	4317	3, 3, 3	52	2	1	1	none

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
