# Peer review of "Pan-Genome-Based Analysis as a Framework for Demarcating Two Closely Related Methanotroph Genera *Methylocystis* and *Methylosinus"

_microorganisms, 2020, doi:10.3390/microorganisms8050768_

Round 1

Reviewer 1 Report

Oshkin et al. use pan-genome analysis to resolve the phylogeny of two methanotrophic genera of the family Methylocystaceae. This article is well crafted and relevant as the phylogeny of those methanotrophic genera are difficult to resolve with the 16S or the pmoA alone, but the author should nevertheless provide some more precision before publishing this manuscript. The figure are especially hard to understand and precious informations are lacking in the legends in order to make the figures comprehensible. Details are below.

minor comments :

Line 26. It would be better to define or explain the concepts of shell and cloud genome before using it for the first time.

Line 64. pmoA should not have a capital p and should be italicised.

Line 127. The color code of blue for Methylosinus and purple for Methlocystis should be indicated in the table legend.

Line 171. The quality of Figure 1 appears bad in the text. It is hard to read the taxon name other than in the Methylosinus cluster. Does the different colors dots with name of systems correspond to the systems from which the different genera were isolated ? Please specify.

Line 174. There are no highlight in purple in the figure.

Line 191. The terms core, soft-core, shell and cloud genomes should be defined before.

Lines 192-193. The separation between shell and cloud genes seems rather arbitrary. What proportions of genome are several vs a few ?

Line 200. Why using two methods here ? The author should explain their rational. Also, the results of the two methods do not match, the author should discuss this discrepancy.

Line 211. The Figure 3 is not clear and there are not enough explanation in the figure legend to understand. What are the different outer rings after the different genomes ? What does the color shading represent for those ? Is it a scale ? Where is the color key of the heatmap ? What represent the lines below the heatmap ? Please give more explanation, the reader should be able to understand the figure simply with the legend.

Line 214. The author use the nomenclature as core, shell and cloud gene while in the figure it says core or singleton, how can the two be reconciled ?

Line 216. But what mean the different shades of purple and blue in the figure ?

Line 229. Again what represent the color key on those heatmaps ?

Line 285. To what correspond the four different networks ? Please specify in the figure legend.

Line 304. Figure 7 is hard to understand. What does the color key of grey shades correspond to ? What are the pale greenish arrow for ?

Line 383. The explanation of what are MCPs should come before.

Reviewer 2 Report

The paper is a follow up paper to report a complete genome of Methylocystis heyeri H2T; and compare this genome to 23 currently available genomes of Methylocystis and Methylosinus species with aim to confirm further that members of these genera form two separate clades.

Please clarify/give definition for “core, shell and cloud gene clusters”

L380 – “Methylocystis try to adapt… Methylosinus move to more favorable environment…” – what environmental factors play role in the adaptation? Does it mean that Methylosinus is not adapted to environment they inhabit? Both genera have flagella/motility related genes – and this is very interesting topic why Methylocystis are not motile? Could it be because of mutation or gene deficiency? Are species of Methylocystis OBBP, BRCS2 and ATCC49242, which have genes encoding flagella, motile?

L397 – 23 genomes were analyzed, authors mentioned these genomes were from 7 characterized and 10 yet-to-be characterized species – could you revise to make clear what about other 6 genomes – where they from characterized or un-characterized species?

L399 Authors mentioned “identification of diverse genome‐encoded traits that determine adaptations to a wide variety of environments” – could they clarify it with regard to these 2 genera

Figure 2 – Colors in the captions and in figure do not completely match.

Figure 5 – What does the color intensity correspond to?

Figure 6 – It is not clear why this figure is presented? Give more description why it is important?
